# Unilateral divorce laws affect women's welfare

**Riccardo Ciacci** [ID]*, **María José Martín Rodrigo, Antonio Núñez Partido**

Department of Economics and Business Management, Universidad Pontificia Comillas, Spain, Madrid

* rciacci@comillas.edu

## Abstract

This paper studies whether unilateral divorce affects women's welfare. Unilateral divorce refers to a divorce regime where each of the spouses can dissolve the marriage unilaterally (i.e. without mutual consent). First, it builds a simple theoretical model that finds that women are better off under unilateral divorce than under mutual consent. Second, it makes use of data from the U.S. between 2003 and 2014 to explore empirically whether unilateral divorce affects the amount of time women devote to three different activities that might be seen as proxies of their level of welfare, such as, housework, leisure and relaxing activities, and personal care. We find causal evidence suggesting that unilateral divorce improves women's welfare. Namely, it reduces housework carried out by women, while it increases their amount of time devoted to leisure and relaxing activities, and personal care. Further results suggest these changes are not due to improvements in gender equality per se. Moreover, we find that the decrease in housework and the surges in leisure and relaxing activities are permanent, whereas the increase in personal care is temporary. These findings are important from a policy perspective to motivate the introduction of unilateral divorce laws.

## Introduction

In the U.S. before 1969, divorce had to be mutually agreed by both marital partners and was granted only on grounds of guilt of misconduct for one of the spouses. As a consequence, consent of the innocent party was necessary for a petition of divorce to be approved. Grounds of guilt of misconduct comprised abandonment, cruelty, incurable mental illness, or adultery. For the reasons set out above, dissolution of marriages broken without misconduct of any spouse was only possible if one of the two parties declared herself or himself guilty. Moreover, since divorce had to be mutually agreed and usually husbands enjoyed higher salaries than wives, there was the belief that if husbands wanted to divorce, they could bribe their wives to get their consent. However, the opposite was less likely since wives could not afford to bribe their partners. A branch of the literature built theoretical models to explore how divorce laws may affect women's welfare and labor participation (see among others [1–6]). Yet, results pointed at different channels, highlighting the importance to address this question empirically without imposing any structure *ex-ante* on the data.a

This paper studies whether unilateral divorce improves women's welfare. Unilateral divorce allows marriage dissolution as long as one of the spouses request it, while, previously with mutual consent divorce, the consensus of both spouses was required. First, this article builds a simple theoretical framework that finds that, under plausible assumptions, women are better

**Data Availability Statement:** All relevant data are within the paper and its Supporting information files.

**Funding:** The authors received no specific funding for this work.

**Competing interests:** The authors have declared that no competing interests exist.

off under unilateral divorce than under mutual consent. Second, it explores empirically whether unilateral divorce affects the amount of time women devote to three different activities that might be seen as proxies of their level of welfare, such as, housework, leisure and relaxing activities, and personal care. We find empirical evidence suggesting that the enforcement of unilateral divorce decreases housework carried out by women, while it boosts their amount of time devoted to leisure and relaxing activities, and personal care. Further results suggest these changes take place after the enforcement of the law. Moreover, we find that the decrease in housework and the surges in leisure and relaxing activities are permanent, whereas the increase in personal care is temporary. Our results are robust to different checks.

This paper contributes to an extensive strand of the literature that evaluates the impact of unilateral divorce laws on divorce rates (see, e.g., [7–10]) and, more generally, on different outcomes (see, inter alia [6, 11–20]). Specifically, this paper is related to a stream of research that finds that unilateral divorce improves different outcomes that appear to be positively correlated with women's welfare, such as, boosts in women's labor supply [20] and reductions in domestic violence, total fertility and prostitution [11, 12, 16, 21, 22].

The contributions of this paper rely on a double methodology. Similar to other papers quoted in this study that rely on both a theoretical model and an empirical analysis (see, inter alia, [6, 9]). First, this manuscript develops a simple theoretical framework where it derives two propositions showing that under plausible assumptions, the introduction of unilateral divorce laws improves wives's welfare. Second, it exploits different empirical techniques linked to econometric analysis to find support for this claim using data from the US between 2003 and 2014.

To the best of our knowledge, previous literature did not answer a similar research question nor it explored the variables used in this paper. Thereby, the contribution of this paper to the literature is twofold. On the one hand, it suggests different measures of women's wellbeing. On the other hand, it finds that unilateral divorce improves women's outcomes for those variables.

The rest of the paper is organised as follows. Section explores the legal and socio-economic background of unilateral divorce in the U.S. Section presents the theoretical model. Section carries out the empirical analysis. Finally, Section discusses results and concludes.

## Legal and socio-economic background

Unilateral divorce laws arose during the second half of the 20th century, simultaneously with other movements such as the civil rights movement or the feminism movement. As a matter of fact, during the seventies, the number of divorces increased due to various social factors, such as, the abolition of laws restricting marriage between people of different ethnicities, the Supreme Court's recognition of marriage as a fundamental right, and the elimination of the "fault divorce" requirement in many states for marriage dissolution.

To this extent, [17] find that the age gap between spouses at the time of marriage progressively narrowed. Namely, the median age for first marriage in 1890 was 22 years old for women and 26 years old for men, while in 2004 it was 26 years old for women and 27 years old for men.

A further important factor in the transformation of the classical family structure, was the introduction of the morning-after pill. In 1961, 41% of married women under 30 years old took the pill [23]. In this age of women's rights liberalization, the Congress approved different laws that facilitated the distribution of the pill to unmarried women. This meant that "by 1976, three-quarters of all unmarried women between 18 and 19 years of age had taken the pill" [17].

It is also key to highlight, the impact of the technological progress in house chores. Inventions such as the dishwasher, dryer or the washing machine considerably reduced the daily time spent on housework [24]. In addition, in some cases these devices became substitutes to tasks that were historically performed by women. As a result, by freeing up the time that was previously devoted to domestic tasks, women could now focus themselves to the development of other activities and interests. Furthermore, since traditionally these activities were carried out by married women, marriage became more attractive to women [17].

Simultaneously, the change in the wage structure, also had an important effect on the classic family structure. Between the 1970s and 1980s in cities where male salary differentials were on the increase, the percentage of young married women declined dramatically [25]. [25] document that cities where the male salary gap rose more rapidly during the seventies and the eighties led to at least one-third of the total decline in marriage rates for women between the ages of 21 and 30. As a consequence, in those places where the male wage gap rose, the opportunity cost of remaining single for women decreased. Hence, many women waited longer to get married. Put it differently, more women decided to postpone the decision to start a family in favor of other priorities, a result which is also consistent with [17].

Last but not least, it is important to consider the change from mutual consent to unilateral consent divorce. The introduction of this law changes drastically the classical family structure and at the same time might be seen as the legislative reflection of many other key changes in society. With the development of a legal framework to protect the spouses for unilateral dissolute their marriage, new opportunities appeared for women's professional development outside of marriage, resulting in an alteration of the classical family structure [24].

In 1969, California became the first U.S. state to completely abolish both the "mutual consent" and "fault divorce" regime requirements with the passage of the California Family Law Act. The fault divorce regime required "misconduct" on the part of one of the spouses either "abandonment", "incurable mental illness", "cruelty" or vadultery". Whereas, the mutual consent regime required mutual agreement between the two spouses to dissolve their marriage. The California Family Law Act also introduced the concept of alimony, the financial support a spouse is ordered to give to his/her ex-spouse. However, if one of the spouses was found guilty of "misconduct" he/she could be punished by losing the right to alimony, child custody or by imposing economic responsibilities [12].

The California Family Law Act started a movement to reform divorce laws in the U.S. In fact, since then various states followed suit, this movement is informally known as *The Divorce Revolution* [18]. *The Divorce Revolution* gathered an apolitical consensus and promoted the introduction of both unilateral and no fault divorce regimess. The former allows marriage dissolution as long as one of the spouses request it, while, previously with mutual consent divorce, the consensus of both spouses was required. The latter eliminates "the test of guilt or innocence of the previous regime" and allowed spouses to invoke divorce for "irreconcilable differences" or "incompatibilities".

A recent strand of the literature explored how unilateral divorce affected different outcomes, such as, domestic violence, homicide, suicide, fertility, children education and prostitution. [11] suggest that unilateral divorce lowers the cost of marital dissolution and, as a result, also dilutes the value of the commitment between the spouses. Likewise, the spouse who seeks to divorce gains bargaining power under the unilateral divorce regime. Using data drawn from the National Vital Statistics of the USA, [11] find that the introduction of unilateral divorce reduces both total and out-of-wedlock fertility.

Coherently with such findings, [16] find that the introduction of unilateral divorce caused a reduction of about 30% in domestic violence, a decline between 8 and 16% in female suicide and a 10% decline in females murdered by their partners. Additionally, [20] finds that

unilateral divorce boosted both married and unmarried labor force participation independently of property division laws. [14] reports the long-term effects of unilateral divorce on children. [14] documents that children who experienced a parental divorce end up with lower education rates and family income; and higher odds of adult suicide. Moreover, these individuals get married at younger ages and separate more often. [12] finds that unilateral divorce reduced female prostitution in the U.S. by 10%. Further empirical results suggest that unilateral divorce makes marriage more attractive to women, leading consequently to an increase in the opportunity cost of female prostitutes.

## Theoretical framework

This section builds a simple theoretical model to study the effect of unilateral divorce laws on womens' welfare, it finds that unilateral divorce leads to higher wives'welfare. Let the sub-index $i = m$ denotes men's variables, while $i = w$ denotes women's variables. Let $w_i$ be the wage for individual $i = m, w$; $m_i$ is a binary variable that measures the *happiness* of the marriage for individual $i = m, w$. Hence, $m_i = 1$ means the marriage is happy, whereas $m_i = 0$ means the marriage is not. Assume a marriage is happy with probability $p_i$ and unhappy with probability $1 - p_i$ where clearly $0 < p_i < 1$.

The binary variable $d_i$ measures whether individual $i$ wants and can obtain a divorce. If a divorce takes place $d_i = 1$, otherwise $d_i = 0$. Under mutual consent divorce: $d_i = 1 - max(m_i, m_j)$, i.e. a divorce occurs if and only if both spouses are in an unhappy marriage (i.e. $m_i = m_j = 0$). Under unilateral consent divorce: $d_i = 1 - min(m_i, m_j)$, i.e. a divorce occurs if one of the spouses is in an unhappy marriage (i.e. $m_i = 0$) [19, 26].

Let the utility function $U_i$ depend on wage $w_i$ and whether an individual can stay in a happy marriage or leave an unhappy marriage $m_i$. Thus, $U_i = u(w_i, |m_i - d_i|)$, where $u(x, y)$ is increasing in each argument. Assume $w_m > w_w$ due to the historical gender wage gap and that under a unilateral divorce regime, if the divorce takes place, wives receive an alimony $a$ where $a \in (0, w_m)$ [6, 9, 15, 18]. Therefore, under unilateral divorce if a divorce takes place wives receive $w_w + a$ and husbands $w_m - a$. In order to measure welfare consider the expected utility function defined as:

$$\mathbb{E}(U_i) = p_i[p_j u(w_i, |m_i - d_i|) + (1 - p_j)u(w_i, |m_i - d_i|)]$$
$$+(1 - p_i)[p_j u(w_i, |m_i - d_i|) + (1 - p_j)u(w_i, |m_i - d_i|)] =$$
$$p_i[p_j u(w_i, 1) + (1 - p_j)u(w_i, |m_i - d_i|)]$$
$$+(1 - p_i)[p_j u(w_i, |m_i - d_i|) + (1 - p_j)u(w_i, 1)]$$

Note that $d_i$ depends on individuals $i$ and $j$ so $u(w_i, |m_i - d_i|)$ cannot be factored out from the brackets. Moreover, since $d_i$ depends on the divorce regime, expected utility $\mathbb{E}(U_i)$ depends on the divorce regime as well.

Finally, to simplify notation denote the expected utility under mutual consent as $\mathbb{E}^m(U_i)$ and under unilateral divorce as $\mathbb{E}^u(U_i)$.

**Proposition 1**. *if* $u(w_w + \bar{a}, 0) \geq u(w_w, 1)$ *for some* $\bar{a} > 0$ *then*:
$\forall a \geq \bar{a} \Rightarrow \mathbb{E}^u(U_w) \geq \mathbb{E}^m(U_w)$

*Proof.* Follows easily substituting in the equations and noting that

$\mathbb{E}^m(U_w) = p_w[p_m u(w_w, 1) + (1 - p_m)u(w_w, 1)] + (1 - p_w)[p_m u(w_w, 0) + (1 - p_m)u(w_w, 1)] \leq$

$p_w[p_m u(w_w, 1) + (1 - p_m)u(w_w + a, 0)] + (1 - p_w)[p_m u(w_w + a, 1) + (1 - p_m)u(w_w + a, 1)] = \mathbb{E}^u(U_w)$

This proposition establishes that if wives'preferences for marriage and divorce can be compensated with an alimony, then there is an alimony high enough such that wives are always

better off under unilateral divorce than mutual consent divorce. However, the threshold value of $\bar{a}$ depends on the probabilities $p_w$ and $p_h$, and so it might be unfeasible.

There is no reason a priori to believe that the probabilities of having a happy marriage should differ across sexes. This consideration introduces Proposition 2.

**Proposition 2**. *If* $p_w = p_h \Rightarrow \mathbb{E}^u(U_w) \geq \mathbb{E}^m(U_w)$

*Proof*. Follows easily substituting in the equations.

This proposition suggests that if it is plausible to assume that the probabilities of success of a marriage are the same across sexes, wives are always better off under unilateral divorce than under a mutual consent regime, regardless of the alimony.

Propositions 1 and 2 suggest that it only suffices to make a plausible assumption: such as that unwanted outcomes in divorce can be compensated monetarily via an alimony or that the probability of happiness in a marriage is the same for men and women, to get the result that unilateral divorce raises wives'welfare.

## Empirical analysis

This section empirically explores the theoretical results found in Section. First, we briefly present the used dataset. Next, we introduce our empirical strategy and comment our results.

### Data

This paper makes use of the American Time Use Survey (hereinafter, ATUS) database from 2003 to 2018. This database is sponsored by the Bureau of Labor Statistics and is carried out by the United States Census Bureau. ATUS keeps track of the amount of time people spend on different activities which range from working and primary activities to leisure. This survey was firstly conducted in 2003 and when this article was written was available till 2018.

In order to proxy women's welfare this paper uses records on the time spent by women on three different activities: housework, leisure and relaxing activities, and personal care. [12] conducts a similar analysis using three similar activities: personal care, sports, and leisure and relaxing activities. Precisely these activities are labelled in the ATUS dataset as: household activities; socializing, relaxing, and leisure activities; and personal care activities.

Such activities are described in the ATUS classification system as:

- **Household activities**: *Household activities are those done by respondents to maintain their households. These include housework; cooking; yard care; pet care; vehicle maintenance and repair; and home maintenance, repair, decoration, and renovation. Food preparation, whether or not reported as done specifically for another household member, is always classified as a household activity, unless the respondent identified it as a volunteer, work, or income-generating activity. For example, "making breakfast for my son" is coded as a household activity, not as childcare. Household management and organizational activities—such as filling out paperwork, balancing a checkbook, or planning a party—also are included in this category. Although all mail and e-mail activities are originally classified in the household activities category during coding, these activities are pulled out of the household activities and included in the composite category Telephone, Mail, and E-mail category in published tables.*

- **Socializing, relaxing and leisure**: *Socializing includes face-to-face social communication with others and hosting or attending parties, receptions, ceremonies, and meetings. Time spent communicating with others using the telephone, mail, or e-mail is not part of this category. Leisure activities include relaxing; playing computer, board, or card games (unless playing with children only); watching television; using a computer or the internet for personal interest; playing or listening to music; reading; writing; and all hobbies. Leisure activities that are active in*

*nature, such as yard games like croquet or horseshoes, are classified under Sports, Exercise, and Recreation. Since 2004, this category captures social activities such as communicating with others and attending parties and meetings; and leisure activities such as relaxing, playing (passive) games (unless playing with children only), watching television, playing or listening to music, reading, writing, and all hobbies. Arts, cultural, and entertainment activities also are coded here, and include attending events or shows related to nature (zoo, arboretum), the arts (galleries, poetry readings), amusement (amusement parks, circus, sightseeing), and performance (plays, ballet). All activities that fall under this category are those done for personal interest or leisure.*

- **Personal care activities**: *Personal care activities include sleeping, bathing, dressing, grooming, health-related self-care, and personal or private activities. Receiving unpaid personal care from others (for example, "my sister put polish on my nails") is also captured in this category. Respondents are not asked who they were with or where they were for personal activities, as such information can be sensitive.*

Using these three activities we want to measure different features of women's welfare. Specifically, we see housework as an unpleasant activity that women carry out disproportionately more often in several countries [27]. Hence, a more balanced division of housework seems to be welfare enhancing for women. There might be the concern that in married couples it might be hard to renegotiate housework activities. Yet, it seems plausible to think that these frictions are less important considering data for women as a whole (i.e. regardless of their marital status). We interpret results connected to leisure and relaxing activties, and personal care as pleasant activities. Hence, we expect unilateral divorce to reduce housework and to boost leisure and personal care.

Table 1 displays descriptive statistics for these three variables depending on the treatment status of the state. Panels A, B, C and D respectively show results for untreated, already treated, treated in the pre-treatment period and treated in the post-treatment period. Comparing Panels A, B and C suggests that there are not observable differences between states, as for these three outcomes, depending on their treatment status. Our sample period spans from 2003 to 2018, there are three states that were treated during those year: Missouri, New Jersey and New York. Footnote of Table 1 displays the month and year in which unilateral divorce became effective in such states. Panel D suggests there are not clear differences in the three outcome variables before and after the effective date of unilateral divorce in treated states. Finally, data on legalization of same sex marriage is drawn from [28]. [28] finds causal evidence that same sex marriage legalization improved employability among same-sex couples.

Our identification assumption relies on the staggered enforcement of unilateral divorce regimes across states. To this extent, it is worth noting that states of the three groups (i.e. untreated, already treated and treated) are comparable to each other and geographically close to each other. Omission of any of these states ought be motivated on lack of comparability, which a priori is unclear in this setting since we simply consider all the U.S. states. Moreover, inclusion of all the states improves precision of the estimates by increasing the number of observations.

## Empirical strategy and results

We consider regression models of the following form:

$$Y_{asy} = \beta Unilateral_{sy} + \mathbf{X}_{asy}\delta + \alpha_y + \alpha_s + \alpha_s * y + \varepsilon_{asy} \tag{1}$$

where *Unilateral*$_{sy}$ is a binary variable taking value 0 if unilateral divorce is not effective in

**Table 1. Summary statistics: Housework, leisure and personal care.**

| | Obs | Mean | Std. Dev | Min | Max |
|---|---|---|---|---|---|
| Panel A: Untreated | | | | | |
| Housework | 18,510 | 135.47 | 127.17 | 0 | 1240 |
| Leisure | 18,510 | 293.95 | 191.16 | 0 | 1434 |
| Personal Care | 18,510 | 588.25 | 136.65 | 0 | 1440 |
| Panel B: Already treated | | | | | |
| Housework | 39,154 | 143.05 | 132.01 | 0 | 1310 |
| Leisure | 39,154 | 280.96 | 185.29 | 0 | 1433 |
| Personal Care | 39,154 | 584.81 | 129.32 | 0 | 1440 |
| Panel C: Treated, pre- period | | | | | |
| Housework | 461 | 143.93 | 132.0783 | 0 | 975 |
| Leisure | 461 | 265.20 | 165.34 | 0 | 1115 |
| Personal Care | 461 | 568.53 | 115.83 | 260 | 1055 |
| Panel D: Treated, post- period | | | | | |
| Housework | 5,530 | 137.47 | 125.12 | 0 | 1365 |
| Leisure | 5,530 | 292.56 | 185.46 | 0 | 1200 |
| Personal Care | 5,530 | 580.83 | 125.47 | 0 | 1425 |

Treated states during sample period, treated years in brackets:

Missouri (September 2009), New Jersey (January 2007), New York (October 2010).

Control states during sample period (i.e. no unilateral divorce without separation requirements):

Arkansas, District of Columbia, Illinois, Lousiana, Maryland, Mississippi North Carolina, Ohio, Pennsylvania South Carolina, Tennessee, Vermont, Virginia.

Already treated states (i.e. unilateral divorce introduced before sample period):

Alabama, Alaska, Arizona, California, Colorado, Connecticut Delaware, Florida, Georgia, Hawaii, Idaho, Indiana, Iowa, Kansas, Kentucky, Maine Massachuetts, Michigan, Minnesota, Montana, Nebraska, Nevada, New Hampshire New Mexico, North Dakota, Oklahoma, Oregon, Rhode Island, Texas, Washington Wisconsin, Wyoming.

state $s$ in year $y$ and taking value 1 when unilateral divorce becomes effective and afterwards. $\alpha_y$, $\alpha_s$ and $\alpha_s{}^*y$ are respectively year fixed effect, state fixed effects and state-year trends. $\mathbf{X_{asy}}$ is a vector of binary controls for women's marital status and cohort (i.e. age) $a$ living in state $s$ in year $y$. Finally, $Y_{asy}$ is one of the three ATUS variables introduced above.

Table 2 presents our main results. In this table, different specifications are provided to check the robustness of such results. With this aim in mind, columns (1) to (5) present results using different sample periods and including different controls. Columns (1), (2) and (3) display results for the sample period 2003–2014 to match the same sample period of [12]. Columns (4), (5) and (6) present the results for the whole sample. Columns (1) and (4) do not include controls $\mathbf{X_{asy}}$. Columns (2) and (5) introduce results adding cohort fixed effects. On top of these controls, columns (3) and (6) add marital status fixed effects. This last specification is Eq (1), our preferred regression model.

Panels A, B and C of Table 2 respectively introduce results for housework, leisure and relaxing activities, and personal care. In line with [12], columns (1), (2) and (3) suggest that women's welfare improved due to unilateral divorce between 2003 and 2014. This result might explain the found decay in prostitution: improving women's welfare, and as a consequence reducing gener inequality might reduce voluntary sex work.

Additionally, as expected, column (6) of Table 2 shows that unilateral divorce is associated to a decay in housework and an increase in both leisure and relaxing activities, and personal care. Using summary statistics of Panel C of Table 1, tells us that unilateral divorce laws are

**Table 2. Regression results: Housework, leisure and personal care.**

|  | (1) | (2) | (3) | (4) | (5) | (6) |
|---|---|---|---|---|---|---|
| Panel A: Housework |  |  |  |  |  |  |
| Unilateral | -10.69*** | -11.59*** | -12.79*** | -11.99*** | -13.22*** | -14.38*** |
|  | (2.418) | (2.424) | (2.356) | (1.921) | (1.989) | (1.948) |
| Panel B: Leisure |  |  |  |  |  |  |
| Unilateral | 37.72*** | 38.26*** | 39.02*** | 23.22*** | 23.15*** | 24.03*** |
|  | (3.329) | (3.283) | (3.318) | (2.575) | (2.653) | (2.611) |
| Panel C: Personal Care |  |  |  |  |  |  |
| Unilateral | 10.88*** | 11.97*** | 12.45*** | 14.97*** | 15.73*** | 16.49*** |
|  | (1.985) | (1.990) | (1.984) | (1.630) | (1.641) | (1.658) |
| Observations | 49,304 | 49,304 | 49,304 | 63,655 | 63,655 | 63,655 |
| Clustered variance at State level |  |  |  |  |  |  |
| State FE | Y | Y | Y | Y | Y | Y |
| State Year Trends | Y | Y | Y | Y | Y | Y |
| Year FE | Y | Y | Y | Y | Y | Y |
| Cohort FE |  | Y | Y |  | Y | Y |
| Marital FE |  |  | Y |  |  | Y |
| Sample | 2014 | 2014 | 2014 | 2018 | 2018 | 2018 |

Clustered standard errors at state level in parentheses

*** p<0.01,

** p<0.05,

* p<0.1

associated to: a decay of about 10% in time spent on housework by women, an increase of about 8% in time spent on leisure and relaxing activities by women and a surge of around 3% in time spent on personal care by women. Comparing results of Table 2 across columns suggests our findings are robust to different model changes.

## Causal estimates

Eq (1) is a difference-in-differences regression model. Given the staggered adoption of unilateral divorce law, this specification seems the most suitable for this setting. In addition, this specfication allows the researcher to explicitly state the assumptions that provide a causal interpretation of the estimates. Under the parallel trends assumption between treated and control states $\hat{\beta}$ measures the effect of unilateral divorce on either housework, leisure and relaxing activities, or personal care. According to such an assumption treated and control states shared parallel trends prior to treatment introduction.

To this extent, there might the doubt that unilateral divorce laws are more likely to be approved and enforced earlier in states where women *historically have* or *gained across years* more bargaining power. This motivates the inclusion of both state fixed effects and state-year trends. Likewise, it might be thought that in certain years women acquired higher bargaining power; this justifies the inclusion of year trends.

It is important to highlight that causal interpretation of the estimates above presented relies on the plausibility of the identification assumption of parallel trends. Nonetheless, such an assumption is untestable. In staggered treatment research designs, as the one considered in this paper, event studies might show key evidence supporting or discarding such an

assumption. Accordingly, Figs 1–3 respectively show the results of running an event study analysis for treated states on the three outcomes considered in this study.

Figs 1–3 report the estimated coefficients for any result, but they report the 95% confidence interval only of statistically significant estimates. In these three figures there is no pre-treatment difference statistically significant, this evidence supports the identification assumption. Put it differently, in these three figures there is no empirical evidence hinting at the possibility that unilateral divorce affected these three outcomes prior to its enforcement. As it seems plausible to assume, finding evidence that the effect took place before the introduction of unilateral divorce law would be evidence discarding that the effect is due to the introduction of such a law. It is worthy to remark that we find no evidence of this sort in any of the three outcomes.

Figs 1–3 suggest that the decrement in housework takes place 1 year after the enforcement of unilateral divorce law and is permanent, the surge in leisure takes place in the year in which unilateral divorce is enforced and is permanent as well. Whereas, the boost in personal care occurs only one year after the introduction of the law and is temporary.

An important recent branch of the literature developed different econometric techiniques that help testing whether difference-in-differences estimates are robust to heterogeneous effects across units and time. This section also unpacks these issues. First, this section provides different empirical checks recently developed that are related to the staggered adoption of the treatment and that test whether our difference-in-differences estimates are robust. Second, this section tackles issues related to the concern that it is not unilateral divorce the cause of the improvement of wives' welfare, but rather, the broad liberalization of women's rights.

There might be concerns about the robustness of our results due to the inclusion of already treated states and to the staggered treatment timing. These two features might even flip the

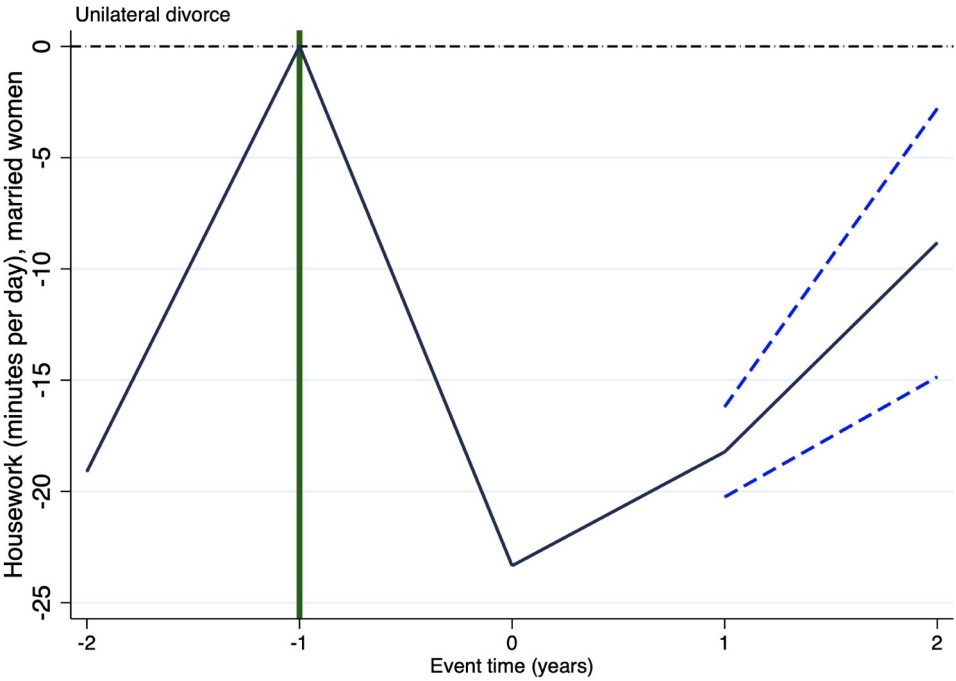

**Fig 1. Event study: Housework.** Notes: This figure shows the estimated coefficients and 95% confidence intervals, for statistically significant estimates, of running an event study of the effect of unilateral divorce on housework. $t = -2$ ($t = 2$) measures the effect of unilateral divorce on the outcome variable 2 years before (after) the enforcement of the law and prior (posterior).

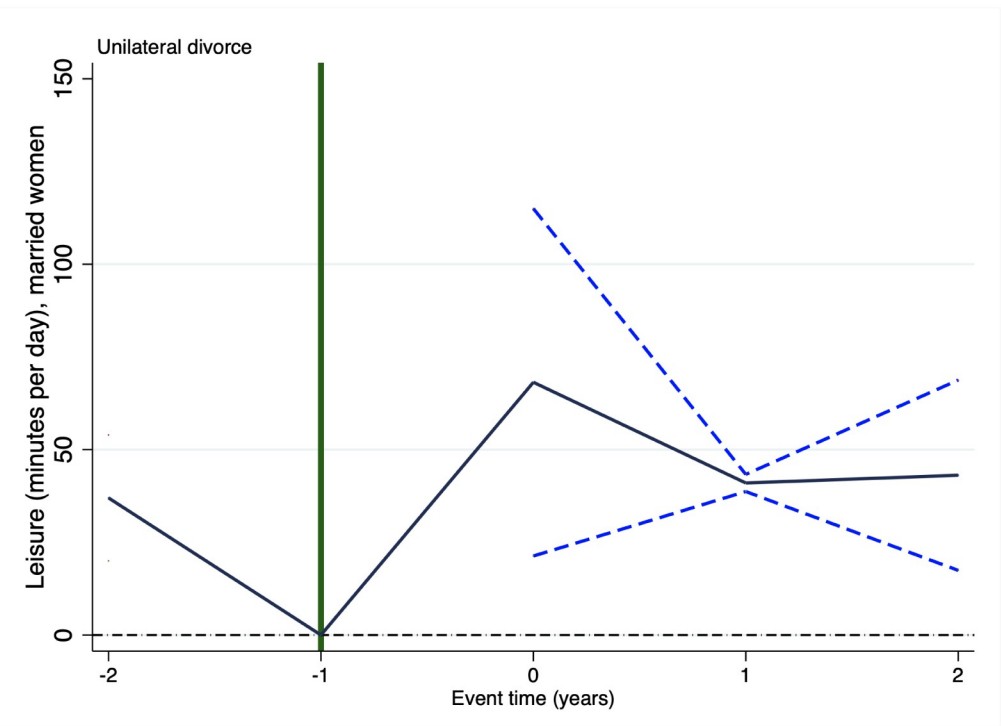

**Fig 2. Event study: Leisure.** Notes: This figure shows the estimated coefficients and 95% confidence intervals, for statistically significant estimates, of running an event study of the effect of unilateral divorce on leisure and relaxing activities. $t = -2$ ($t = 2$) measures the effect of unilateral divorce on the outcome variable 2 years before (after) the enforcement of the law and prior (posterior).

sign of our estimates. On this regard, Table 4 in S1 Appendix performs a number of checks suggested by [29] to address such issues. This table finds that our estimates are robust across specifications. Namely, column (1) considers the simplest specification possible (i.e. baseline specification, as column (4) in Table 1), while columns (2), (3) and (4) respectively consider the main specification but using either only treated states, only treated and untreated states, or only treated and already treated states. Columns (5) adds age-marital status interacted fixed effects. Columns (6) and (7) respectively use either only unit specific trends as controls or outcomes detrended by group status specific pre-trends.

It is reassuring to find that for the three outcome variable considered results are stable across specifications. Furthermore, Table 4 in S1 Appendix also displays the difference between each estimate and the baseline and the main specification one. Such differences take, in general, low values.

[29] shows that the difference-in-differences estimator is a weighted average of estimators across comparison groups. Such groups might even carry negative weights. This issue leads to two potential concerns. First, it might be that estimators differ considerably from the weighted average. Second, negative weights might even lead to flip the sign of the weighted average with respect to estimators across comparison groups. To address this issue [29] suggests a decomposition method. To apply this method the dataset needs not to have repeated values across time periods. Hence, in this setting it is necessary to collapse the dataset at cohort and marital level.

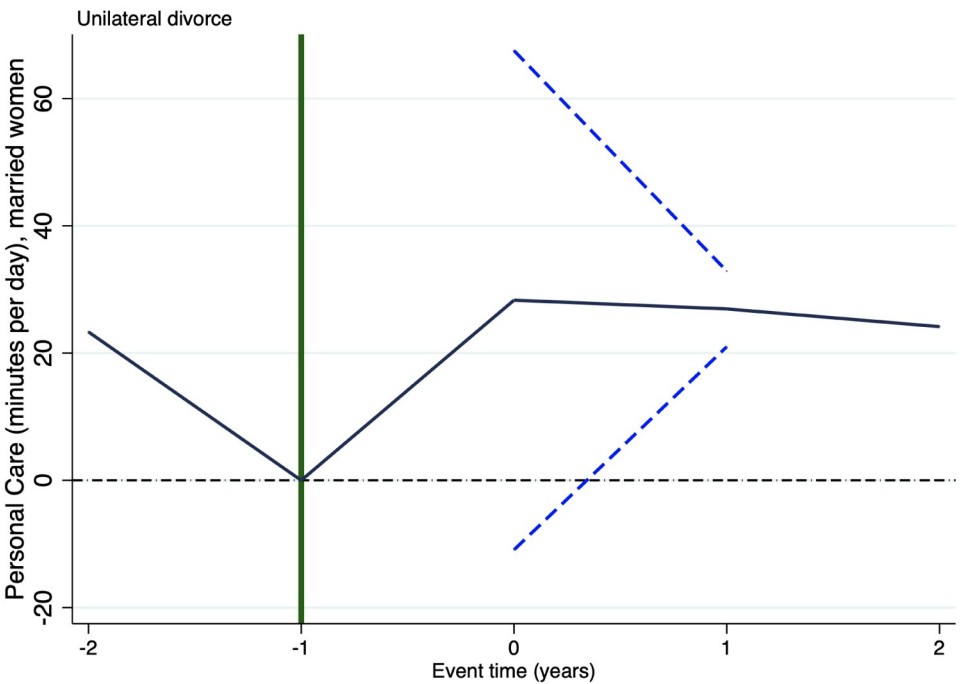

**Fig 3. Event study: Personal care.** Notes: This figure shows the estimated coefficients and 95% confidence intervals, for statistically significant estimates, of running an event study of the effect of unilateral divorce on personal care. $t = -2$ ($t = 2$) measures the effect of unilateral divorce on the outcome variable 2 years before (after) the enforcement of the law and prior (posterior).

The first line of Table 5 in S1 Appendix presents the results for such regressions. Demanding readers might see such regressions as further robustness checks. The number of observations clearly is lower than the ones of the main specification. Moreover, this regression, with respect to our main one, might only include year fixed effects, state fixed effects and state year trends. Cohort and marital fixed effects cannot be used since the dataset is collapsed to average those two variables. It is reassuring to find that results are statistically equal to our main estimates (i.e. column (6) of Table 2).

Table 5 in S1 Appendix also shows the results of decomposing the difference-in-differences estimates. In our setting the main comparisons are across comparability of already treated vs treated and untreated vs treated. The former carries a weight close to 76%, the latter the remaining 24%. It is encouraging to find that for each of the three outcome variables estimates across groups are similar in size to the corresponding main estimate. Thereby, in our setting weights and decomposition of the difference-in-differences estimator offer further support in favor of our findings.

There might be concerns that the treatment effects of our difference-in-differences estimates might be heterogeneous across age or marital status of women, since we analyze—given the sample period of available data—late introducers of unilateral divorce law. To address these concerns, Appendix Table 6 in S1 Appendix shows results of different checks developed in [30]. This table shows that even if 46% of the difference-in-differences weights are negative the correlation between those weights and age and marital status is small in size and statistically equal to zero. This suggests that effects of unilateral divorce are not heterogeneous along those two dimensions.

There might be concerns regarding the fact that during our sample period only three states were treated. We addressed these concerns showing summary statistics of our outcome variables in Table 1. In here, we address these concerns using randomization inference. Namely, it might be thought that due to the low number of treated states during our sample period, such states are not comparable to control states. Making use of randomization inference we can compute *how likely it is to find our results by chance.*

Figs 4–6 show the results of randomizing unilateral divorce (i.e. our treatment variable), with 100 permutations and stratifying at Census Bureau regions, for our three outcomes. As expected, in these three figures we observe that the average effect estimated by randomization inference is zero. Furthermore, the area below the interesection of the main coefficient (i.e. the red line) with the density can be seen as the probability to find our results by chance. Since in none of the figures they intersect each other these figures tell us that the probability to find our results by chance is close to zero.

There might be the doubt that our findings are not due to the entry into force of unilateral divorce but to the expansion of gender equality as a whole. To this extent, it might be thought that our results are an effect of any policy linked to women's rights liberalization. To address this issue we replace our treatment variable with the effective date of introduction of Same Sex Marriages (hereinafter, SSM). SSM seems to be the best choice for this analysis since SSM legalization took place during the sample period considered in this paper. Indeed, 26 states legalized SSM between 2004 and 2007. Further information on this regard is discussed in [28]. Table 3 shows the results for such regressions. We find that SSM did not affect any of our three outcome variables. Indeed, for every specification considered our estimates are not statistically different from zero and their magnitude is small (i.e. the lack of significance does not seem to be driven by large standard errors).

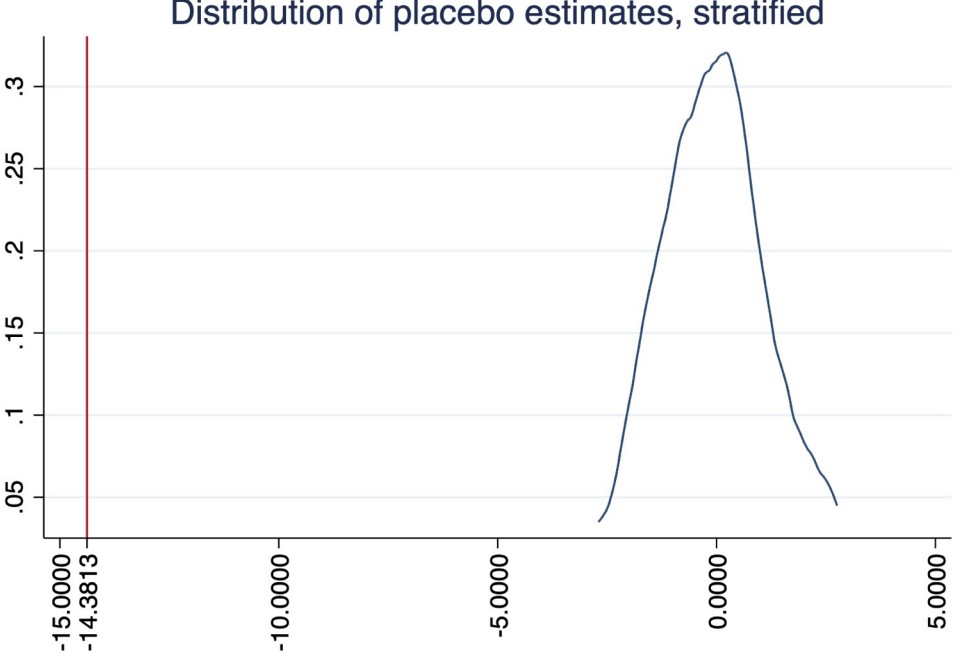

**Fig 4. Randomization inference: Housework.**

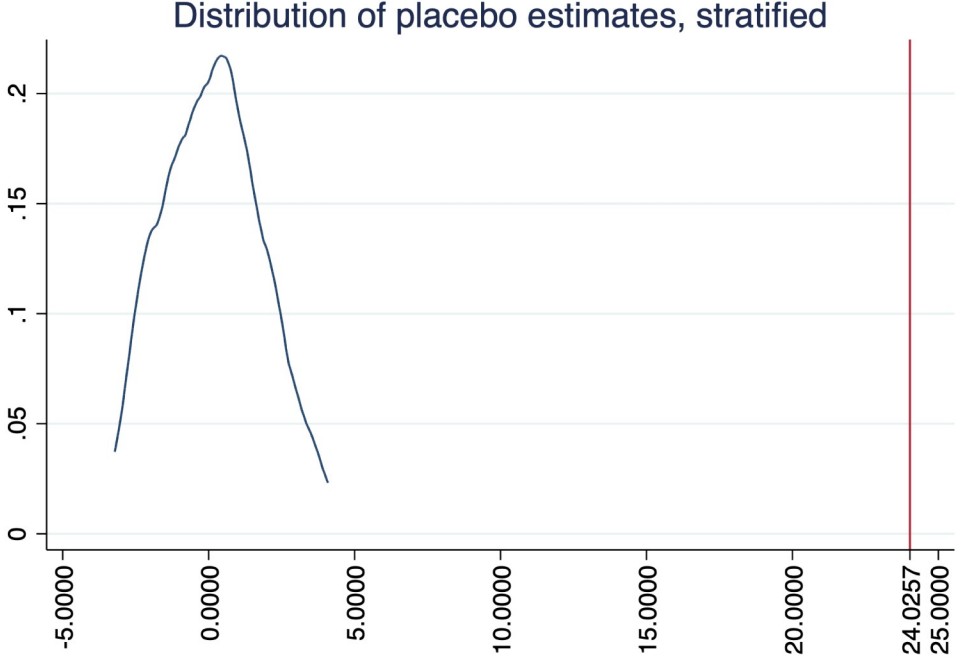

**Fig 5. Randomization inference: Leisure and relaxing activities.**

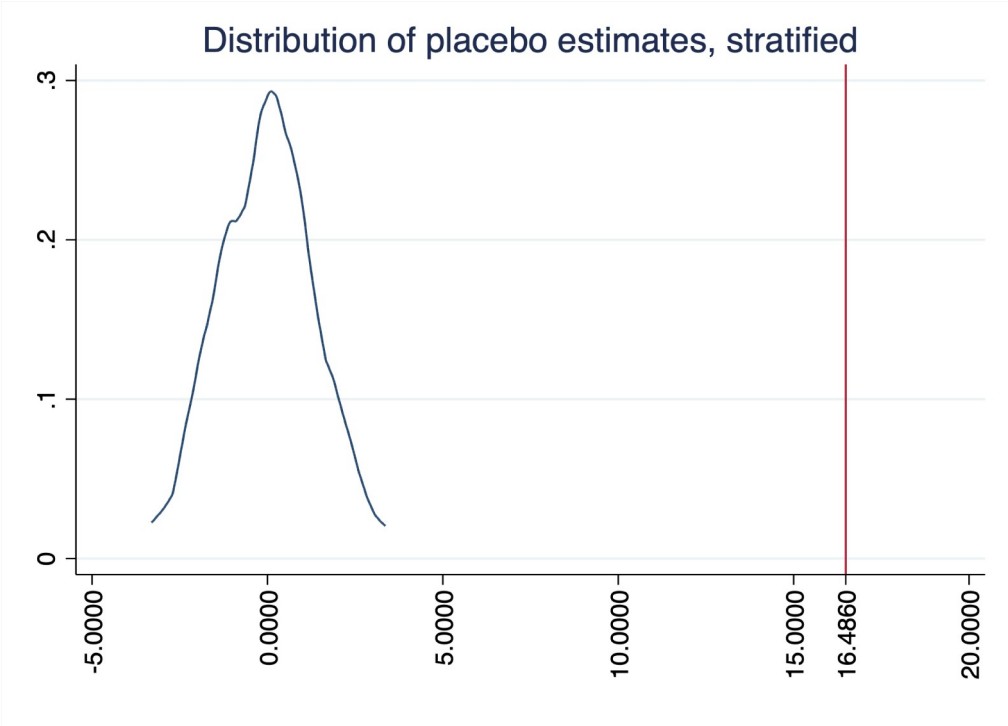

**Fig 6. Randomization inference: Personal care.**

**Table 3. SSM regression results: Housework, leisure and personal care.**

|  | (1) | (2) | (3) | (4) | (5) | (6) |
|---|---|---|---|---|---|---|
| Panel A: Housework | | | | | | |
| SSM | -1.719 | -2.326 | -1.971 | -0.706 | -1.145 | -1.066 |
|  | (3.242) | (3.368) | (3.412) | (2.204) | (2.258) | (2.262) |
| Panel B: Leisure | | | | | | |
| SSM | -4.865 | -5.790 | -5.696 | -4.051 | -5.792 | -5.559 |
|  | (4.803) | (4.814) | (4.770) | (3.800) | (3.956) | (3.936) |
| Panel C: Personal Care | | | | | | |
| SSM | -0.178 | -0.325 | -0.446 | 0.466 | 0.0820 | 0.123 |
|  | (2.825) | (2.816) | (2.876) | (3.039) | (2.976) | (3.017) |
| Observations | 49,304 | 49,304 | 49,304 | 63,655 | 63,655 | 63,655 |
| Clustered variance at State level | Y | Y | Y | Y | Y | Y |
| State FE | Y | Y | Y | Y | Y | Y |
| State Year Trends | Y | Y | Y | Y | Y | Y |
| Year FE | Y | Y | Y | Y | Y | Y |
| Cohort FE |  | Y | Y |  | Y | Y |
| Marital FE |  |  | Y |  |  | Y |
| Sample | 2014 | 2014 | 2014 | 2018 | 2018 | 2018 |

Clustered standard errors at state level in parentheses

*** $p<0.01$,

** $p<0.05$,

* $p<0.1$

To further address this concern. Table 7 in S1 Appendix presents the results of our main analysis adding also SSM as a control variable. If women's rights liberalization affected women's welfare, the inclusion of SSM should affect our results. Table 7 in S1 Appendix shows that our results are unaffected by SSM inclusion and that the estimated coefficient of SSM is low in value and statistically insignificant across regression models. These findings support the notion that unilateral results is the main driver between the found effects.

All in all, these results suggest that unilateral divorce improved women's welfare. Results seem robust across specifications, suggest the parallel trend identification assumption is plausible and that they cannot be explained by the low number of treated states (i.e. 3) during our sample period (2003 to 2018).

## Concluding remarks

This paper studies whether unilateral divorce improves women's welfare. To this extent, this manuscript contributes to both the theoretical and empirical literature providing supportive evidence that unilateral divorce improves women's wellbeing across different outcomes.

First, it builds a simple theoretical model. According to this model, it is sufficient to assume either that unwanted outcomes in divorce can be compensated monetarily via an alimony or that the probability of happiness in a marriage is the same for men and women, to obtain the result that women are better off under unilateral divorce than under mutual consent.

Second, it explores empirically whether unilateral divorce affects the amount of time women devote to three different activities that might be seen as proxies of their level of welfare, such as, housework, leisure and relaxing activities, and personal care. We find empirical

evidence suggesting that the enforcement of unilateral divorce decreases housework carried out by women, while it boosts their amount of time devoted to leisure and relaxing activities, and personal care. Moreover, our analysis finds that these results are robust to a number of checks recently developed in the difference-in-differences literature [29, 30].

Further results suggest these changes take place after the enforcement of the law and are not due to changes in gender equality per se, but unilateral divorce specifically. Moreover, we find that the decrease in housework and the boosts in leisure and relaxing activities are permanent, whereas the increase in personal care is temporary. Our results are robust to different checks.

These three activities measure different dimensions of women's welfare. There is evidence in the literature that a more balanced housework schedule would enhance welfare for women [27]. Furthermore, while household chores might be harder to re-bargain in married couples this is not the case for leisure and relaxing and for personal care activities. A priori there is no reason to believe that these effect could be different for women's in different situations (e.g. mothers or different marital status).

All in all, this paper relates to a strand of the literature that finds that unilateral divorce improves different outcomes that appear to be positively correlated with women's welfare, such as, boosts in women's labor supply [20] and reductions in domestic violence, total fertility and prostitution [11, 12, 16]. To this extent, the contribution of this paper to the literature is twofold. First, it suggests usage of different variables to measure women's wellbeing. Second, it offers supportive evidence that unilateral divorce improves women's wellbeing. It is worth mentioning that a limitation of this analysis is that it relies on divorce law changes in a country where unilateral divorce was accepted apolitically, results might be different for countries with a different political situation surrounding this law [12]. These findings are important from a policy perspective to motivate legalization of unilateral divorce and be aware of the impacts of such a law.

## Supporting information

**S1 Appendix.**
(PDF)

## Acknowledgments

This article is drawn from an undergrad thesis entitled *Did unilateral divorce law improve women's welfare?* (original title: *Las Leyes de Divorcio Unilateral. ¿¿Han mejorado el bienestar de las mujeres?*) directed by Professor Riccardo Ciacci and written by Rodrigo García Flores. In comparison with this thesis, theoretical results have been generalized and empirical analysis advanced. All remaining errors are our own.

## Author Contributions

**Conceptualization:** Riccardo Ciacci.

**Data curation:** Riccardo Ciacci.

**Funding acquisition:** María José Martín Rodrigo, Antonio Núñez Partido.

**Methodology:** Riccardo Ciacci.

**Writing – original draft:** Riccardo Ciacci.

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
