## [Decision Letter · Decision Letter 0]

20 Apr 2023

PONE-D-22-29658Comments on the link between unilateral divorce and women's welfarePLOS ONE

Dear Dr. Ciacci,

Thank you for submitting your manuscript to PLOS ONE. After careful consideration, we feel that it has merit but does not fully meet PLOS ONE’s publication criteria as it currently stands. Therefore, we invite you to submit a revised version of the manuscript that addresses the points raised during the review process.

Please see the comments of two reviewers below. We hope that these are useful in order to strengthen the manuscript. In addition to these comments, I would suggest changing the title to include some details about the methodology employed in the study, in order that readers don't mistake this for a review article or opinion piece.

We look forward to receiving your revised manuscript.

Kind regards,

Hanna Landenmark

Staff Editor

PLOS ONE

Journal Requirements:

Additional Editor Comments (if provided):

Reviewers' comments:

Reviewer's Responses to Questions

**Comments to the Author**

1. Is the manuscript technically sound, and do the data support the conclusions?

Reviewer #1: Yes

Reviewer #2: Partly

2. Has the statistical analysis been performed appropriately and rigorously? 

Reviewer #1: Yes

Reviewer #2: I Don't Know

3. Have the authors made all data underlying the findings in their manuscript fully available?

Reviewer #1: Yes

Reviewer #2: No

4. Is the manuscript presented in an intelligible fashion and written in standard English?

Reviewer #1: Yes

Reviewer #2: Yes

5. Review Comments to the Author

Reviewer #1: The paper presents a very interesting study of effect of unilateral divorces on time-use of women from a well-being perspective. Though the time-use pattern may be subjective to many more social and economic factors, the current paper and research questions on its own also presents a intriguing and novel perspective to the literature that evaluates the impact of unilateral divorce laws on divorce rates and various outcomes.

Reviewer #2: General comments

1. The paper is well structured in general, with a well-written theoretical framework and research question background.

2. At places, I didn’t find enough explanations for your decisions in analysis and arguments.

3. There wasn’t enough evidence in your introduction or conclusion that you’re adding something new to this literature. It would be nice to explicitly mention that at the end of the introduction and in the conclusion, you can mention in a paragraph that, concerning the current literature, what new knowledge you are adding.

4. Some more robustness tests might be required to prove the reliability of your analysis. More on that in the next section.

Specific Comments

1. The abstract didn’t carry enough information about the country of the study, the methods you used and the time period. It will be nice to mention that briefly. Also, add a sentence about unilateral divorce in the abstract, as people outside the US haven’t heard of this term before.

2. Maybe add a few sentences about the definition of unilateral divorce in the second paragraph of your introduction, as people outside the US haven’t heard of this term before.

3. It would be nice to explicitly mention at the end of the introduction specifically what you’re contributing to the literature and if the outcomes you’re looking at have been looked at before. If yes, then what’d they find?

4. Clearly specify which states are part of which panels in Table 1. It will be easier to understand with years and names of states in Panel B, C and D.

5. Describe briefly why do you think the states that you have chosen as control states are good controls for this period? How’re they similar to the treated states regarding population size, characteristics, and race? I see that you put state fixed effects in the model, but still I’m curious about the rationale for selecting these states as controls?

6. The results in your table 2 is in terms of minutes that they spend additionally or less for a certain activity, right? But it will be interesting to see it in terms of the per cent change in these activities, which will be effect size related to the mean size of these activities.

7. Was the Event study conducted for all the treated states together or particularly for a state? Please mention.

8. I don’t understand the rationale for your event study figures very well. Why are the CIs available for only times 1 &2, and why there is a line at t=-1? Please explain that a bit more in your figure caption or results.

9. Appendix Table A.1 performs a number of checks suggested by Goodman-Bacon (2021) to address issues due to the staggered treatment timing and that we use already treated states in the control group. Can you briefly mention these suggested checks, as an average reader might not be aware of those?

10. You can also do a decomposition analysis of your DID estimate using the “bacondecomp” package in Stata or R, which will decompose the effect size and weights contributed by each of the four comparisons that you have shown to your final estimate.

11. Finally, your conclusion doesn’t discuss the literature sufficiently and places your contribution in that literature. I would advise writing a paragraph on the current state of literature and what knowledge you added to that in your conclusion. Also, mention some limitations of your analysis regarding assumptions and generalizability of your findings.

6. PLOS authors have the option to publish the peer review history of their article (what does this mean?). If published, this will include your full peer review and any attached files.

Reviewer #1: No

Reviewer #2: **Yes: **Rishabh Tyagi

---

## [Author Response · Author response to Decision Letter 0]

13 May 2023

In here I paste the replies to the comments of the referees that might be found also in the response to referees file

Reviewers' comments:

Reviewer's Responses to Questions

Comments to the Author

1. Is the manuscript technically sound, and do the data support the conclusions?

Reviewer #1: Yes

Reviewer #2: Partly

2. Has the statistical analysis been performed appropriately and rigorously? 

Reviewer #1: Yes

Reviewer #2: I Don't Know

3. Have the authors made all data underlying the findings in their manuscript fully available?

Reviewer #1: Yes

Reviewer #2: No

Data are attached to the current submission of the R&R version of the article.

4. Is the manuscript presented in an intelligible fashion and written in standard English?

Reviewer #1: Yes

Reviewer #2: Yes

5. Review Comments to the Author

Reviewer #1: The paper presents a very interesting study of effect of unilateral divorces on time-use of women from a well-being perspective. Though the time-use pattern may be subjective to many more social and economic factors, the current paper and research questions on its own also presents a intriguing and novel perspective to the literature that evaluates the impact of unilateral divorce laws on divorce rates and various outcomes.

Thanks.

Reviewer #2: General comments

1. The paper is well structured in general, with a well-written theoretical framework and research question background.

Thanks.

2. At places, I didn’t find enough explanations for your decisions in analysis and arguments.

Please let me know where this occurs and I can change them as you see fit.

3. There wasn’t enough evidence in your introduction or conclusion that you’re adding something new to this literature. It would be nice to explicitly mention that at the end of the introduction and in the conclusion, you can mention in a paragraph that, concerning the current literature, what new knowledge you are adding.

I just added this in both the introduction and the conclusion. I can change them further as you see convenient.

4. Some more robustness tests might be required to prove the reliability of your analysis. More on that in the next section.

Ok, I will perform every test you suggest to give further credibility to the study.

Specific Comments

1. The abstract didn’t carry enough information about the country of the study, the methods you used and the time period. It will be nice to mention that briefly. Also, add a sentence about unilateral divorce in the abstract, as people outside the US haven’t heard of this term before.

Many thanks for this comment that might help readers to better understand the scope and results of the paper. I have added each of the suggestions you name.

2. Maybe add a few sentences about the definition of unilateral divorce in the second paragraph of your introduction, as people outside the US haven’t heard of this term before.

Done. Please let me know if I can modify it further.

3. It would be nice to explicitly mention at the end of the introduction specifically what you’re contributing to the literature and if the outcomes you’re looking at have been looked at before. If yes, then what’d they find?

I added a new paragraph tackling this comment and a previous comment about the contribution. I can further change it as you consider appropriate.

4. Clearly specify which states are part of which panels in Table 1. It will be easier to understand with years and names of states in Panel B, C and D.

I added also already treated states to the list below Table 1. They were the only set of states I did not include since the other two categories implicitly defined those states. Yet, in this way it is much clearer for readers. Many thanks for the comment.

5. Describe briefly why do you think the states that you have chosen as control states are good controls for this period? How’re they similar to the treated states regarding population size, characteristics, and race? I see that you put state fixed effects in the model, but still I’m curious about the rationale for selecting these states as controls?

I added a whole paragraph addressing this issue before the Empirical strategy and results subsection. I can further edit it as you see fit.

6. The results in your table 2 is in terms of minutes that they spend additionally or less for a certain activity, right? But it will be interesting to see it in terms of the per cent change in these activities, which will be effect size related to the mean size of these activities.

I analyze this issue in page 11. Namely, I state “Using summary statistics of Panel C of Table 1, tells us that unilateral divorce laws are associated to: a decay of about 10% in time spent on housework by women, an increase of about 8% in time spent on leisure and relaxing activities by women and increment of around 3% in time spent on personal care by women.” I can change it as you consider appropriate.

7. Was the Event study conducted for all the treated states together or particularly for a state? Please mention.

All the treated states together. I clarified this issue in the new version of the article.

8. I don’t understand the rationale for your event study figures very well. Why are the CIs available for only times 1 &2, and why there is a line at t=-1? Please explain that a bit more in your figure caption or results.

As for the vertical line, it is because t=-1 is the last period before the introduction of the treatment. Only confidence intervals different from zero are presented so that the figure can be zoomed in. Whenever confidence intervals are not presented we know that they are large enough to be statistically equal to zero. In the notes of the figures it is written “This figure shows the estimated coefficients and 95% confidence intervals, for statistically significant estimates,” I can edit it further with the wording you see fit.

9. Appendix Table A.1 performs a number of checks suggested by Goodman-Bacon (2021) to address issues due to the staggered treatment timing and that we use already treated states in the control group. Can you briefly mention these suggested checks, as an average reader might not be aware of those?

Sure, I modified a paragraph in page 17 to accommodate this comment. Please let me know if you would like me to carry out further changes in this direction.

10. You can also do a decomposition analysis of your DID estimate using the “bacondecomp” package in Stata or R, which will decompose the effect size and weights contributed by each of the four comparisons that you have shown to your final estimate.

I did not use this command because, as far as I know, it cannot be used in this specific setting since it relies on the xtreg command that does not allow repeated time values within the panel as it is the case of this setting. Indeed, this setting explores data across age groups, state and years (subscripts a, s,y in the empirical specification). Hence, in each state-year cell there are multiple age groups and even different marital status. Then, xtreg cannot be used and consequently the command bacondecomp does not work either. 

However, to address your comment, I have added an extra check. I collapse the data at both cohort (i.e., age) and marital status level to avoid having repeated values for each state and year. Then, I run a regression using the fixed effects of the main regression (which might be seen as a further robustness check) and for such a regression I can decompose the difference-in-differences estimator using the “bacondecomp” method. I present these results in Table A.2 and add a brand-new discussion about it. Please let me know if this solves your concerns, if not I can add any further check you see fit.

11. Finally, your conclusion doesn’t discuss the literature sufficiently and places your contribution in that literature. I would advise writing a paragraph on the current state of literature and what knowledge you added to that in your conclusion. Also, mention some limitations of your analysis regarding assumptions and generalizability of your findings.

Thanks for this comment. I have edited the last paragraph of the Conclusion section to accommodate these comments. I can further modify it as you consider best.

---

## [Decision Letter · Decision Letter 1]

1 Jun 2023

PONE-D-22-29658R1Unilateral divorce laws affect women's welfarePLOS ONE

Dear Dr. Ciacci,

Thank you for submitting your manuscript to PLOS ONE. After careful consideration, we feel that it has merit but does not fully meet PLOS ONE’s publication criteria as it currently stands. Therefore, we invite you to submit a revised version of the manuscript that addresses the points raised during the review process.

We look forward to receiving your revised manuscript.

Kind regards,

Ajoke Basirat Akinola, Ph.D.

Academic Editor

PLOS ONE

Journal Requirements:

Additional Editor Comments:

The article is a good one, but it needs some research methodology re-structuring. Thereby making the article strongly scientific, systematically outlining the methodology section and the subsections. More clarity is needed on the study design adopted, and justification for the same with references.

Sir, according to PLOS ONE journal publication criteria, I suggest you kindly modify this manuscript as it's already published online in 2021 as below 

https://repositorio.comillas.edu/xmlui/bitstream/handle/11531/55100/Comments%20on%20the%20link%20between%20unilateral%20divorce%20and%20womens%20welfare.pdf?sequence=1

Reviewers' comments:

Reviewer's Responses to Questions

**Comments to the Author**

1. If the authors have adequately addressed your comments raised in a previous round of review and you feel that this manuscript is now acceptable for publication, you may indicate that here to bypass the “Comments to the Author” section, enter your conflict of interest statement in the “Confidential to Editor” section, and submit your "Accept" recommendation.

Reviewer #2: All comments have been addressed

Reviewer #3: All comments have been addressed

2. Is the manuscript technically sound, and do the data support the conclusions?

Reviewer #2: Yes

Reviewer #3: Yes

3. Has the statistical analysis been performed appropriately and rigorously? 

Reviewer #2: Yes

Reviewer #3: Yes

4. Have the authors made all data underlying the findings in their manuscript fully available?

Reviewer #2: Yes

Reviewer #3: Yes

5. Is the manuscript presented in an intelligible fashion and written in standard English?

Reviewer #2: Yes

Reviewer #3: Yes

6. Review Comments to the Author

Reviewer #2: (No Response)

Reviewer #3: (No Response)

7. PLOS authors have the option to publish the peer review history of their article (what does this mean?). If published, this will include your full peer review and any attached files.

Reviewer #2: No

Reviewer #3: **Yes: **Dr. M Murtaza Farkhan MD, MPH

---

## [Author Response · Author response to Decision Letter 1]

6 Jun 2023

Please check the document labelled as "Response to Reviewers", yet I copy paste part of it here:

Journal Requirements:

Many thanks for this comment. In the last revision, I was asked to re-write the whole article using the Plos One template. To this extent, I have done my best to adapt the article and reference list to this template Yet, I have not retracted any reference if you have any doubt about any specific citation, please let me know. On this regard, I have also sdouble checked each citation in the new version of the paper I am submitting with respect to the first submission and found in the latest version there are the papers of the first version plus some new articles added during this revision process.

Additional Editor Comments:

The article is a good one, but it needs some research methodology re-structuring. Thereby making the article strongly scientific, systematically outlining the methodology section and the subsections. More clarity is needed on the study design adopted, and justification for the same with references.

Sir, according to PLOS ONE journal publication criteria, I suggest you kindly modify this manuscript as it's already published online in 2021 as below 

https://repositorio.comillas.edu/xmlui/bitstream/handle/11531/55100/Comments%20on%20the%20link%20between%20unilateral%20divorce%20and%20womens%20welfare.pdf?sequence=1

Thanks for this comment. I have rewritten different sections of the article with the objective to restructure the article and outline and highlight the methodology and study design chosen at each stage. I have colored those lines in red in the revised version. Furthermore, I can add any other specific explanation on the methodology used that you consider appropriate. If this is the case please let me know specifically which section/s need further editing and what questions about the methodology I should address. 

As for the publication criteria, the link you provide is a university repository of the working papers each professor writes. It is not a published article but a working paper version. Indeed, for this very same reason, you can find there other articles I wrote and published on journals. However, if you want it removed from that repository for publication criteria I can ask and get it removed.

Many thanks for your comments and suggestions to improve the paper.

Best regards,

---

## [Editor Report · Decision Letter 2]

23 Jun 2023

PONE-D-22-29658R2Unilateral divorce laws affect women's welfarePLOS ONE

Dear Dr. Ciacci,

Thank you for submitting your manuscript to PLOS ONE. After careful consideration, we feel that it has merit but does not fully meet PLOS ONE’s publication criteria as it currently stands. Therefore, we invite you to submit a revised version of the manuscript that addresses the points raised during the review process.ACADEMIC EDITOR: Thank you so much your response. This paper is really interesting and I really appreciate the author. However, there are two major types of research designs. Primary and Secondary research designs. Which one of these have you employed? Or is it combination of both. Sir, it's stated that this article employed a double methodology. Kindly support with references from literatures and justify the double methodology. In my opinion e.g. one could say mixed method of research design and there are lots of articles/ references to support mixed method of design, which should be stated. Similarly in this study explaining the reasons and importance of using the double methodology is paramount. I find it a little confusing, when there isn't enough justification for this. Please research and find articles, books e.tc. supporting, the same that we can relate, if not available, which I suppose should be, since that's what the article is built on. Still if not available, kindly use mixed method of research and support with available and similar literatures (the simple theoretical model built should be supported with citations, the idea must have been gotten from somewhere). This is necessary to enhance research integrity, knowledge and knowledge transfer. It's the beauty and purpose of research. Secondly, kindly go through the PLOS one criteria thoroughly. You must not have published the article elsewhere, whether in part or whole or on any university server as working paper. Please retract such , with proof accordingly.For Lab, Study and Registered Report Protocols: These article types are not expected to include results but may include pilot data. 

We look forward to receiving your revised manuscript.

Kind regards,

Ajoke Basirat Akinola, Ph.D.

Academic Editor

PLOS ONE

Journal Requirements:

Additional Editor Comments (if provided):

Thank you so much your response. This paper is really interesting and I really appreciate the author. However, there are two major types of research designs. Primary and Secondary research designs. Which one of these have you employed? Or is it combination of both.

Sir, it's stated that this article employed a double methodology. Kindly support with references from literatures and justify the double methodology. In my opinion e.g. one could say mixed method of research design and there are lots of articles/ references to support mixed method of design, which should be stated. Similarly in this study explaining the reasons and importance of using the double methodology is paramount. I find it a little confusing, when there isn't enough justification for this.

Please research and find articles, books e.tc. supporting, the same that we can relate, if not available, which I suppose should be, since that's what the article is built on. Still if not available, kindly use mixed method of research and support with available and similar literatures (the simple theoretical model built should be supported with citations, the idea must have been gotten from somewhere). This is necessary to enhance research integrity, knowledge and knowledge transfer. It's the beauty and purpose of research.

Secondly, kindly go through the PLOS one criteria thoroughly. You must not have published the article elsewhere, whether in part or whole or on any university server as working paper.

I suggest you to kindly retract such , with proof accordingly.

Thank you.
---

## [Author Response · Author response to Decision Letter 2]

26 Jun 2023

Dear Editor,

I have addressed each comment and provided a reply (in red). Some of them are also addressed in the new version of the paper. Please let me know if you consider appropriate to change further any aspect of the article. Many thanks for your time and help. 

Thank you so much your response. This paper is really interesting and I really appreciate the author. 

Many thanks for your compliments and comments.

However, there are two major types of research designs. Primary and Secondary research designs. Which one of these have you employed? Or is it combination of both. 

It is a combination of both since this paper builds a mathematical model of the problem (this framework does not rely on any data source so it would be closer to primary design) and on top of that, this paper also provides an empirical analysis using different data sources (which would clearly fall in the secondary research design category)

Sir, it's stated that this article employed a double methodology. Kindly support with references from literatures and justify the double methodology. In my opinion e.g. one could say mixed method of research design and there are lots of articles/ references to support mixed method of design, which should be stated. Similarly in this study explaining the reasons and importance of using the double methodology is paramount. I find it a little confusing, when there isn't enough justification for this. Please research and find articles, books e.tc. supporting, the same that we can relate, if not available, which I suppose should be, since that's what the article is built on. Still if not available, kindly use mixed method of research and support with available and similar literatures (the simple theoretical model built should be supported with citations, the idea must have been gotten from somewhere). This is necessary to enhance research integrity, knowledge and knowledge transfer. It's the beauty and purpose of research. 

Sure, please find detailed and highlighted changes in the new version of the manuscript. To this extent I have added a footnote quoting other papers that in economics also provide both theoretical and empirical analysis in the same article and I have also added more specific citations to the assumptions of the model so that the most demanding readers might find out from where they are drawn. 

Please do not hesitate to contact me if you think it would be appropriate to add more information/citations on this regard. In economics it is usual to either build a mathematical model (which in economics we call theory) or to use data to study the issue (which in economics is also known as empirics). 

Secondly, kindly go through the PLOS one criteria thoroughly. You must not have published the article elsewhere, whether in part or whole or on any university server as working paper. Please retract such , with proof accordingly.For Lab, Study and Registered Report Protocols: These article types are not expected to include results but may include pilot data. 

This is not a published version of the study: it differs from the current version of the article, and it is a working paper version deposited in the repository of my university. Yet, I have sent an email to remove it from the afore-mentioned repository. If you want, I can let you know as soon as it is removed.

---

## [Editor Report · Decision Letter 3]

13 Jul 2023

Unilateral divorce laws affect women's welfare

PONE-D-22-29658R3

Dear Dr. Ciacci,

We’re pleased to inform you that your manuscript has been judged scientifically suitable for publication and will be formally accepted for publication once it meets all outstanding technical requirements.

Kind regards,

Ajoke Basirat Akinola, Ph.D.

Academic Editor

PLOS ONE
---

## [Editor Report · Acceptance letter]

25 Sep 2023

PONE-D-22-29658R3 

Unilateral divorce laws affect women's welfare 

Dear Dr. Ciacci:

I'm pleased to inform you that your manuscript has been deemed suitable for publication in PLOS ONE. Congratulations! Your manuscript is now with our production department. 

Kind regards, 

on behalf of

Dr Ajoke Basirat Akinola 

Academic Editor

PLOS ONE